# Clinical Relevance of a 16-Gene Pharmacogenetic Panel Test for Medication Management in a Cohort of 135 Patients

**DOI:** 10.3390/jcm10153200

**Published:** 2021-07-21

**Authors:** David F. Niedrig, Ali Rahmany, Kai Heib, Karl-Dietrich Hatz, Katja Ludin, Andrea M. Burden, Markus Béchir, Andreas Serra, Stefan Russmann

**Affiliations:** 1Drugsafety.ch, 8703 Kusnacht, Switzerland; david.niedrig@hirslanden.ch (D.F.N.); rahmanya@student.ethz.ch (A.R.); 2Hospital Pharmacy, Clinic Hirslanden Zurich, 8032 Zurich, Switzerland; 3Swiss Federal Institute of Technology Zurich (ETHZ), 8093 Zurich, Switzerland; andrea.burden@pharma.ethz.ch; 4INTLAB AG, 8707 Uetikon am See, Switzerland; kai@intlab.online (K.H.); dietrich@intlab.online (K.-D.H.); 5Labor Risch, Molecular Genetics, 3097 Berne, Switzerland; katja.ludin@mailbox.org; 6Center for Internal Medicine, Clinic Hirslanden Aarau, 5001 Aarau, Switzerland; markus.bechir@zim.ch; 7Institute of Internal Medicine and Nephrology, Clinic Hirslanden Zurich, 8032 Zurich, Switzerland; andreas.serramph@hirslanden.ch

**Keywords:** pharmacotherapy, pharmacogenetics, genetic panel tests, clinical relevance, CYP450, SONOGEN XP

## Abstract

There is a growing number of evidence-based indications for pharmacogenetic (PGx) testing. We aimed to evaluate clinical relevance of a 16-gene panel test for PGx-guided pharmacotherapy. In an observational cohort study, we included subjects tested with a PGx panel for variants of *ABCB1*, *COMT*, *CYP1A2*, *CYP2B6*, *CYP3A4*, *CYP3A5*, *CYP2C9*, *CYP2C19*, *CYP2D6*, *CYP4F2*, *DPYD*, *OPRM1*, *POR*, *SLCO1B1*, *TPMT* and *VKORC1*. PGx-guided pharmacotherapy management was supported by the PGx expert system SONOGEN XP. The primary study outcome was PGx-based changes and recommendations regarding current and potential future medication. PGx-testing was triggered by specific drug–gene pairs in 102 subjects, and by screening in 33. Based on PharmGKB expert guidelines we identified at least one “actionable” variant in all 135 (100%) tested patients. Drugs that triggered PGx-testing were clopidogrel in 60, tamoxifen in 15, polypsychopharmacotherapy in 9, opioids in 7, and other in 11 patients. Among those, PGx variants resulted in clinical recommendations to change PGx-triggering drugs in 33 (32.4%), and other current pharmacotherapy in 23 (22.5%). Additional costs of panel vs. single gene tests are moderate, and the efficiency of PGx panel testing challenges traditional cost-benefit calculations for single drug–gene pairs. However, PGx-guided pharmacotherapy requires specialized expert consultations with interdisciplinary collaborations.

## 1. Introduction

Pharmacogenetics is the study of variability in drug responses associated with genetic differences amongst individuals. Drugs for which such variability in their effects has been linked to genetic polymorphisms are also referred to as pharmacogenetic (PGx) drugs [1]. Today, there is a growing list of PGx drugs, but the question of clinical relevance and implications of PGx test results for individual patients poses the next challenge. A widely accepted classification of the relevance of PGx testing for specific drug–gene pairs has been established by the Pharmacogenomics Knowledgebase (PharmGKB) [2]. The three PharmGKB categories with the highest level of evidence and clinical relevance for PGx-testing are termed “required”, “recommended” and “actionable”. Information from PharmGKB is publicly available, continuously updated and based on expert opinions, published research studies, and PGx information from official Summary of Product Characteristics (SmPCs).

Until now, only few PGx drug–gene pairs fall into PharmGKB’s “required” category based on the establishment of a very high attributable risk for (formerly) idiosyncratic, life-threatening adverse drug reactions (ADR) or lack of therapeutic efficacy and therefore a high predictive value of a detected PGx variant. For example, the association of severe skin reactions caused by abacavir and carbamazepine with genetic variants that code for human leucocyte antigens (HLA) fall into that category. After the establishment of sufficient evidence this information is now included in the labels of corresponding drugs, and PGx testing is mandatory before their first administration [3]. For drugs such as the immunosuppressant azathioprine, PGx testing is not mandatory but classified as “recommended” to determine an effective and yet safe starting dose [4]. Other drug–gene pairs are currently only classified as “actionable”, sometimes in spite of a growing body of evidence on the strength of a clinically relevant association. Other factors such as lower costs and widespread availability of PGx-testing may further challenge their classification and promote a general recommendation of preemptive PGx testing for more drug–gene pairs in the future. Examples include prodrugs such as the platelet inhibitor clopidogrel, or tamoxifen for the secondary prevention of breast cancer [5,6,7].

Not only does PGx testing promise to improve efficacy and safety outcomes for patients, it could also lead to overall savings in health care costs due to more efficient patient management strategies. Particularly preemptive PGx testing with multigene panels may be a promising approach for the identification of clinically relevant variants [8]. If they are used in a high number of subjects, costs of PGx testing may decrease considerably and therefore have a major impact on weighing costs vs. benefits.

Despite many potential benefits, the implementation of PGx testing in clinical practice remains a slow process, particularly outside academic institutions. Challenges include limited and sometimes controversial evidence with regard to improved clinical outcomes for many drug–gene pairs [9], discrepancies between guidelines from PGx expert groups vs. different medical specialty associations [7,10,11], reaction time of regulatory authorities regarding the implementation of new PGx evidence, and limited reimbursement of the costs for PGx testing [12,13]. Furthermore, even if a valid PGx test is performed, it may be challenging to find an expert who can interpret its findings and manage pharmacotherapy within a patient’s individual clinical context [14]. Clinical PGx experts must consider not only interactions for one or several drug–gene pairs, but also many other relevant cofactors such as age, comorbidities, comedication and patients’ personal perceptions of risks and benefits.

The utility of PGx as a guiding tool for pharmacotherapy in clinical practice is subject to ongoing studies and controversial debates, and data on the implementation of PGx services in routine clinical practice and subsequent PGx-based changes in medication management is limited. Therefore, the present study describes our experience from the implementation and interpretation of a PGx panel test, and its relevance for the management of current and future pharmacotherapy in individual patients.

## 2. Materials and Methods

### 2.1. Study Design and Ethical Approval

We conducted an observational cohort study that evaluated the results of a 16-gene PGx panel test and their implementation for personalized pharmacotherapy. The primary outcome of the study was the proportion of patients where PGx panel testing had clinically relevant management implications for current or potential future medication.

The study protocol was reviewed and approved by the local ethics board (EKNZ project ID 2020-00565), and all included patients had signed informed consents for PGx testing and scientific use of their health data.

### 2.2. Study Population and Procedures

An overview of the study procedures is presented in Figure 1. We included all subjects who underwent PGx testing with a 16-gene PGx panel between June 2018 and June 2020 through clinical pharmacology services at two Swiss tertiary care hospitals and associated outpatient clinics, i.e., there was no selection of patients presented in this study. The reason for PGx testing was either a specific drug–gene pair relating to current or planned pharmacotherapy, or a request for preemptive PGx screening. For all subjects the indication for PGx testing was first evaluated by a senior clinical pharmacologist (SR), including a consultation and review of all medical diagnoses and pharmacotherapy. If the indication for PGx testing was confirmed, venous blood samples were obtained using EDTA containing Vacutainers. After receipt of PGx test results and automated reports from the SONOGEN XP expert system, the clinical pharmacologist and a senior clinical pharmacist (DN) evaluated all available information and wrote a comprehensive report for each tested subject. The report included personalized PGx-based management recommendations for the attention of patients and treating physicians. If the clinical pharmacologist was in charge of the patient’s therapy, he would also be able to directly change the medication. Patients also received a summary of the PGx profile in a credit card format (Appendix A). If necessary, there was another follow-up consultation with a personal discussion of all results and adjustments of pharmacotherapy.

### 2.3. Genetic Analysis

DNA extraction and PGx analyses were performed by Labor Risch molecular genetics laboratory, Bern-Liebefeld, Switzerland. DNA was extracted using the QIAsymphony^®^ DSP DNA Mini Kit according to manufacturer’s instructions. The isolated DNA was subsequently amplified by means of the iPLEX^®^ assay which consists of multiplex-PCR, shrimp alkaline phosphatase (SAP) reaction and iPLEX^®^ primer extension. The modified products were then separated using the MassARRAY^®^ MALDI-TOF (Matrix-Assisted Laser Desorption Ionization-Time Of Flight) System (PGx 74 with an additional customized multi-PCR mix) by Agena Bioscience (Hamburg, Germany). The analysis of variants included SNPs (single nucleotide polymorphisms) of the following genes: *ABCB1*, *COMT*, *CYP1A2*, *CYP2B6*, *CYP3A4*, *CYP3A5*, *CYP2C9*, *CYP2C19*, *CYP2D6*, *CYP4F2*, *DPYD*, *OPRM1*, *POR*, *SLCO1B1*, *TPMT* and *VKORC1*. Analysis of *CYP2D6* also included determination of copy number variations (CNV). A list of the tested SNPs for each gene is provided in Appendix A.

### 2.4. PGx Expert System

Results of molecular genetics analyses were forwarded to SONOGEN and further processed by its XP expert system. The SONOGEN XP expert system (www.sonogen.eu; latest access date 25 June 2021) provides an interpretation of identified variants of the 16 tested genes and clinical management recommendations for drug–gene variant pairs that are based on its proprietary knowledge database and decision support algorithms. Patients are categorized into metabolizer phenotypes by means of established star allele nomenclature and current guidelines. The phenotypes for the individual genes were assigned according to standardized nomenclature whenever available from the following sources: *ABCB1* [15], *COMT* [16,17], *CYP1A2* [16], *CYP2B6* [16], *CYP2C9* [18,19], *CYP2C19* [16,20,21], *CYP2D6* [22], *CYP3A4* [16], *CYP3A5* [16,23], *CYP4F2* [16], *DPYD* [24], *OPRM1* [16], *POR* [16], *SLCO1B1* [16], *TPMT* [16], *VKORC1* [16]. The SONOGEN XP (latest available version: 1.9.0-0) system generates automated recommendations for current and potential future pharmacotherapy based on pharmacogenetic phenotypes and the classification of their clinical relevance according to PharmGKB (https://www.pharmgkb.org; latest access date 25 June 2021), including variant annotations according to PharmGKB guidelines, as well as other available guidelines from CPIC (https://cpicpgx.org; latest access date 25 June 2021) and DPWG (https://upgx.eu/guidelines; latest access date 25 June 2021). If there are differences in classifications among labels from different countries SONOGEN XP conservatively uses the highest classification. Sonogen XP is a registered and certified medical product classified as a “system for clinical decision support with a focus on pharmacogenetics”. The status has been certified according to EN ISO 13485:2016 by the Swiss Association for Quality and Management Systems (SQS). A sample report in three different available versions is provided as Appendix A.

### 2.5. Retrospective Documentation and Validation

For the retrospective data analysis and validation as part of this study, the clinical pharmacologist (SR), the clinical pharmacist (DN) and a pharmacist in training (AR) reviewed all available original medical records, referral letters, pharmacotherapy prescriptions and laboratory results. Patient characteristics and clinical factors including current pharmacotherapy, laboratory results and medical history were extracted and compiled in a study database. Comedications were also categorized according to their potential for moderate or strong inhibition of cytochrome P450 enzymes CYP2C19 and CYP2D6 according to the MediQ-database (www.mediq.ch; latest access date 25 June 2021), and these inhibitors are presented in Appendix A.

All clinical recommendations from the reports were validated and categorized as appropriate. First, in patients where a specific drug–gene pair was the indication of PGx testing, we documented if the test result of the related gene led to a recommendation to change therapy with the drug that triggered PGx testing. Second, current comedication and results for all 16 genes of the PGx panel were analyzed for any additional clinically relevant drug–gene interactions. Third, for all subjects including those with a screening indication, we documented if any PGx variants were detected that related to a drug–gene pair with “actionable”, “recommended” or “required” classification according to PharmGKB. Such variants were presented in our PGx reports as potentially relevant for future medication and further discussed in the individual clinical context of tested subjects.

Drug-gene pairs, their classification of clinical relevance according to PharmGKB, and the assignment of genotypes to according phenotypes are presented in Appendix A.

### 2.6. Data Analysis

Data analysis was descriptive with stratification and presentation of results in tables as appropriate. Data management, analyses and creation of figures were performed with STATA MP Version 15.1 (STATA Corporation, College Station, TX, USA).

## 3. Results

### 3.1. Characteristics of the Study Population

We included 135 patients that had undergone testing with the 16-gene PGx panel between June 2018 and June 2020 (Figure 1). Patient characteristics are presented in Table 1, including a stratification over drug-specific indication vs. screening. Compared to 33 subjects with a screening indication, the 102 patients with a drug-specific indication for PGx testing were older (median 70 vs. 58 years) and took a higher number of drugs (median 6 vs. 3). The three most frequent drug-specific indications for PGx-testing were therapy with clopidogrel (*n* = 60), tamoxifen (*n* = 15) and polypsychopharmacotherapy (*n* = 9). Medications in the tested population were predominantly related to cardiovascular diseases, but we also observed frequent use of analgesics, antidepressants, antidiabetics and benzodiazepines.

Furthermore, drug–gene interactions may be particularly relevant in the presence of additional drug–drug interactions that affect the same metabolic pathway, or in case of impaired renal function. It is therefore of interest that 19.3% of the study population took inhibitors of CYP2D6 and 8.2% of CYP2C19, and that 14.1% had an eGFR below 60 mL/min.

### 3.2. Pharmacogenetic Variants and Their Clinical Relevance for Current Medication

Phenotypes of the 16 tested genes were derived from the identified PGx variants, and their frequencies in the study population are presented in Figure 2. Table 2 presents an overview of the tested genes, drugs that are affected by these variants along with their corresponding PharmGKB classification, as well as the frequency of these variants in our study population. A detailed listing of drug–gene pairs and their classification of clinical relevance according to PharmGKB is presented in Appendix A.

The 16-gene PGx panel detected genetic variants, i.e., non-wild-type genes, in 3.7% (for *DPYD*) to 80.0% (for *ABCB1*) of all patients. *CYP2D6*, *CYP2C19*, *CYP2C9* and *TPMT* variants are of particular interest because they relate to drugs where PGx testing is classified as required or recommended. Phenotype variants were detected for *CYP2D6* in 49.3%, *CYP2C19* in 54.1%, *CYP2C9* in 34.1% and *TPMT* in 6.7% of the study population. Of note, Table 2 provides the numbers and proportions of all patients with non-wild-type variants, but not all variants necessarily have the same classification for all listed drugs. E.g., the number of subjects with CYP2C19 variants in Table 2 refers to IM, PM as well as to RM and UM phenotypes, but for clopidogrel only the IM and PM phenotypes are “actionable”.

It is also of particular interest, that three patients (2.2%) had a CYP2C19 IM or PM phenotype and in addition a current prescription for a CYP2C19 inhibitor. For CYP2D6, there were even 18 patients (13.3%) with an IM or PM phenotype and an additional current prescription for a CYP2D6 inhibitor.

Therefore, Table 3 presents a detailed analysis for each drug that triggered PGx-testing including the number of patients with related genetic variants. The additional columns present an analysis of the clinical relevance of those variants. First, we present the number of patients where SONOGEN XP recommends to consider a change of the drug that triggered PGx testing. Second, we present the number of patients where the subsequent clinical pharmacology expert evaluation recommended a change of the triggering drug. Third, we present the number of patients where the 16-gene PGx panel identified additional drug–gene variant interactions in their current comedication.

Overall, among 102 patients with a drug-specific indication for PGx testing, actionable variants for the triggering drugs were identified in 36 patients (35.3%) according to SONOGEN XP, and after clinical expert evaluation including further patient-specific factors recommendations to change PGx-triggering drugs were actually issued in 33 patients (32.4%). The majority of these recommendations (19 patients) referred to current therapy with clopidogrel.

Furthermore, the 16-gene PGx panel identified genetic variants that related to the current comedication and led to “coincident” additional clinical recommendations to adjust comedication in 23 out of 102 patients (22.5%) with a drug-specific indication for PGx testing, and in 3 out of 33 patients (9.1%) with a screening indication. Details of PGx-based recommendations on comedication are presented in Appendix A.

### 3.3. Pharmacogenetic Variants and Their Clinical Relevance for Potential Future Medication

The frequencies of patients with a given number of identified PGx variants of different PharmGKB classifications and according recommendations to adjust potential future pharmacotherapy are presented in Figure 3 and Table 4. The 16-gene panel identified at least one “actionable”, “recommended” or ”required” variant in 100% of the tested patients, and in 74.1% we found two or more concomitant “actionable” variants. The prevalence of the highly relevant “recommended” and “required” variants was lower. Still, 73.3% had one, and another 6.7% even two “recommended” variants, 38.5% one “required” variant, and 86.7% of all patients had at least one “recommended” or “required” variant.

As shown in Table 4, the median number of alerts regarding clinically relevant PGx variants for potential future medication was five according to SONOGEN XP. Our reports provided a listing of those recommendations as an attachment, but the actual personalized expert assessments highlighted only those with the highest clinical relevance, hence the median number of recommendations in our personalized clinical reports was only three and therefore lower.

## 4. Discussion

This study describes our experience from the implementation of a 16-gene PGx panel in routine clinical practice with a focus on clinical relevance. The 16-gene PGx panel test was able to detect variants that are clinically relevant according to the PharmGKB classification in 100% of tested patients. More important, results of PGx testing led to an actual change of medication or specific recommendations to do so in a high proportion of the tested patients [25]. These adjustments of current medication and specific recommendations regarding potential future medication were supported by a PGx expert system and implemented through personalized clinical pharmacology consultations.

Overall, frequencies of PGx variants shown in Figure 2 are in agreement with previous studies in Caucasian populations [26,27,28]. The detection rate of 100% for at least actionable variants is an expected finding for a 16-gene PGx panel if one considers that in a previous study even a panel with only five genes had a reported detection rate of 99% [27]. Detection rates are typically based on the PharmGKB classification of clinical relevance, which may be considered as the current gold standard for publicly available PGx knowledge bases. SONOGEN XP further enhances PGx clinical decision support through additional reviews of other knowledgebases, thorough review of the original literature, collaborations with external experts, and an array of separate reports for different purposes. These range from concise reports written for patients, over specific therapeutic recommendations for prescribing physicians, to extensive summaries for experts of ten and more pages including references to original research publications. The very high detection rate of PGx panel tests for variants that are classified as “required”, “recommended” or “actionable” support the use of such multigene PGx panels with the automated interpretation from expert systems for preemptive testing with the ultimate goal to improve efficacy of pharmacotherapy, and to reduce adverse reactions and costs [27,29]. At the same time, it should also be noted that variants of *ABCB1*, *COMT*, *CYP3A4*, *OPRM1* and *POR* are currently included in the used PGx panel, but in accordance with current PGx guidelines we did not consider those as clinically relevant in any of our patients. The composition of the used PGx panel may therefore be subject to future adjustments depending on evolving evidence.

The experience reported in our study also looks beyond PGx panel tests with automated clinical decision support for PGx-based pharmacotherapy and their merely theoretical impact on pharmacotherapy. Whereas Table 2 lists a large number of PGx drugs for the identified PGx variants including some that are hardly ever used (e.g., pimozide or atazanavir), Table 3 provides a real-life insight into the prevalence of specific drugs plus relevant PGx variants that required a change of therapy in our patients. Our patients with a specific indication for PGx testing had a median number of six concomitant drugs. We provided personalized clinical pharmacology consultations and issued personalized expert recommendations to adjust therapy with the PGx-triggering drug, current concomitant medication and potential future medication. We recommended, or if the clinical pharmacologist was directly involved in patient care, directly changed the PGx-triggering drug in 32.4%, and any other concomitant medication as a “bycatch”, in 22.5% of patients based on PGx panel results. This high value supports the clinical relevance of PGx panels for actual clinical decision making and, to our knowledge, has not been investigated in this way before. Because additional costs of panel vs. single gene tests are moderate and likely to further decrease with advancing technology and widespread use, these findings further support the cost-efficiency of PGx panel testing and provide an alternative view at traditional cost-benefit calculations based on single drug–gene pairs.

However, a closer look also reveals that PGx-based management of pharmacotherapy in real-life clinical practice is a complex process, and that the standardized PharmGKB classification can be highly heterogeneous within the same class. For example, PGx testing for clopidogrel and tamoxifen is merely classified as “actionable” according to PharmGKB. But the lack of efficacy associated with the tested PGx variants is potentially lethal, and, based on a review of the latest evidence, PGx expert guidelines, and our own clinical experience, we conclude that PGx testing indeed makes an important contribution to clinical decisions related to those frequently prescribed drugs and can even improve patient compliance [5,6,7,25,30,31]. On the other hand, one must realize that for many other drugs in spite of established statistically significant associations most PGx variants do not have a high predictive value for efficacy or adverse reactions of a drug in individual patients. Rather, they act as one of several factors with complex and often poorly understood interactions, and their effect may be best described by a causative pie model [32]. Accordingly, our clinical experience from PGx-supported clinical decision making reminded us that PGx decision support algorithms are helpful, but that they do not comprehensively capture the complexity of (shared) clinical decision making. As shown in Table 1, we identified a considerable number of patients with comedication inhibiting CYP2C19 or CYP2D6, or renal impairment, and our therapeutic decisions considered all those factors and their interactions with PGx variants, as well as alternative therapeutic options. Indeed, the number of new drugs where the SmPC includes information on PGx variants is steadily increasing. For example, prescription of siponimod (Mayzent^®^) requires preemptive *CYP2C9* PGx testing, and the prescribing information of brexpiprazole (Rexulti^®^) provides dosing recommendations that consider both, PGx variants as well as concomitant therapy with inhibitors of CYP2D6 or CYP3A4. Indeed, among our patients we identified 21 patients with a CYPC19 or CYP2D6 slow metabolizer phenotype and the concomitant prescription of an inhibitor of the corresponding enzyme, which is expected to increase the clinical relevance of those pharmacogenetic variants. Even for drugs that have been marketed for a long time, postmarketing studies may identify previously unknown relevant PGx variants [33]. Therefore, we expect a growing demand for PGx testing outside academic centers with integrated expert consulting by clinical pharmacologists, clinical pharmacists and experts from the respective specialties in the near future.

Some limitations of our study should also be addressed. Our study population was preselected, partially through physicians that referred patients for specific drug–gene indications, and partially through “mere” screening indications. Characteristics of our patients are therefore transparently presented in Table 1, and they may be different in other institutions that offer PGx services. Although our recommendations are a critical appraisal of clinical relevance, we were not able to conduct a larger study with longitudinal follow-up in order to evaluate outcomes of our PGx-based recommendations. These must be addressed in prospective large controlled studies for specific PGx-guided therapy [5,31]. Nevertheless, we were able to perform a separate analysis for our PGx consultations in patients with clopidogrel therapy, and our results are indeed in line with those studies [25]. Another limitation concerns the used 16-gene panel itself. Due to technical reasons this panel did not include relevant *HLA* variants associated with severe adverse reactions towards carbamazepine or abacavir [34,35], but from a medical point of view this would certainly be desirable.

## 5. Conclusions

Our study demonstrates the value of PGx panel testing in routine clinical practice and the valuable contribution of a PGx clinical decision support system. Additional costs of panel vs. single gene tests are moderate and in case of frequent use they can be further reduced through scaling effects. Therefore, the efficiency of PGx panel testing challenges traditional cost–benefit calculations based on single drug–gene pairs. However, a closer look also reveals that truly personalized pharmacogenetic medication management will not achieve its full potential without individual patient consultations where additional factors and individual weighing of risks vs. benefits and pharmacotherapeutic as well non-pharmacotherapeutic care are considered. Limited availability of experts and specialized clinics may become a bottle neck for the implementation of PGx-guided pharmacotherapy that will require additional resources, which is a challenge but also an opportunity and responsibility for clinical pharmacology and clinical pharmacy services to seek direct patient contact and involvement in PGx-guided medication management.

## Figures and Tables

**Figure 1 jcm-10-03200-f001:**
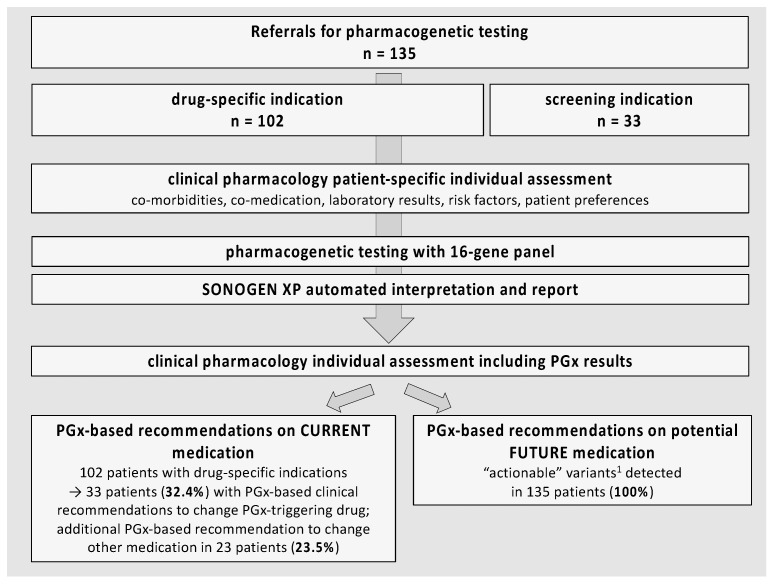
Study population and flowchart. ^1^ Formally classified as “actionable” according to SONOGEN XP based on PharmGKB guidelines.

**Figure 2 jcm-10-03200-f002:**
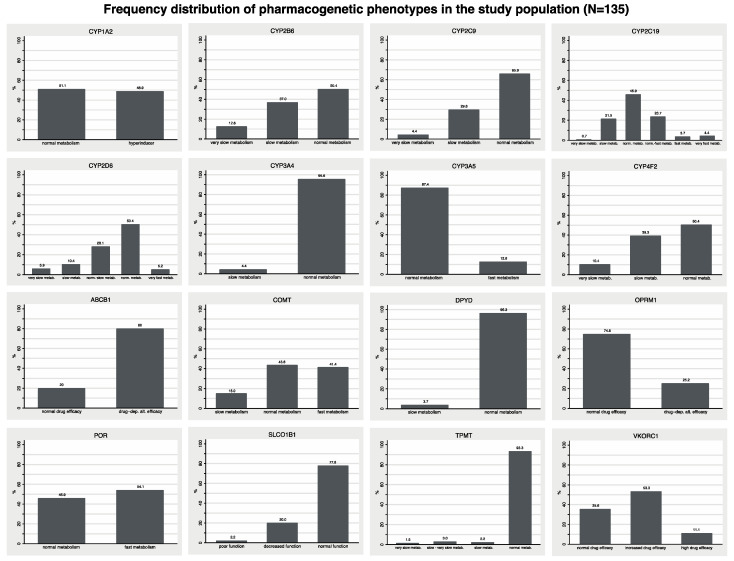
Frequency distribution of pharmacogenetic phenotypes in the study population. Phenotypes for the individual genes were assigned according to the latest available standardized nomenclature (see methods Section 2.4, including references for each genotype/phenotype).

**Figure 3 jcm-10-03200-f003:**
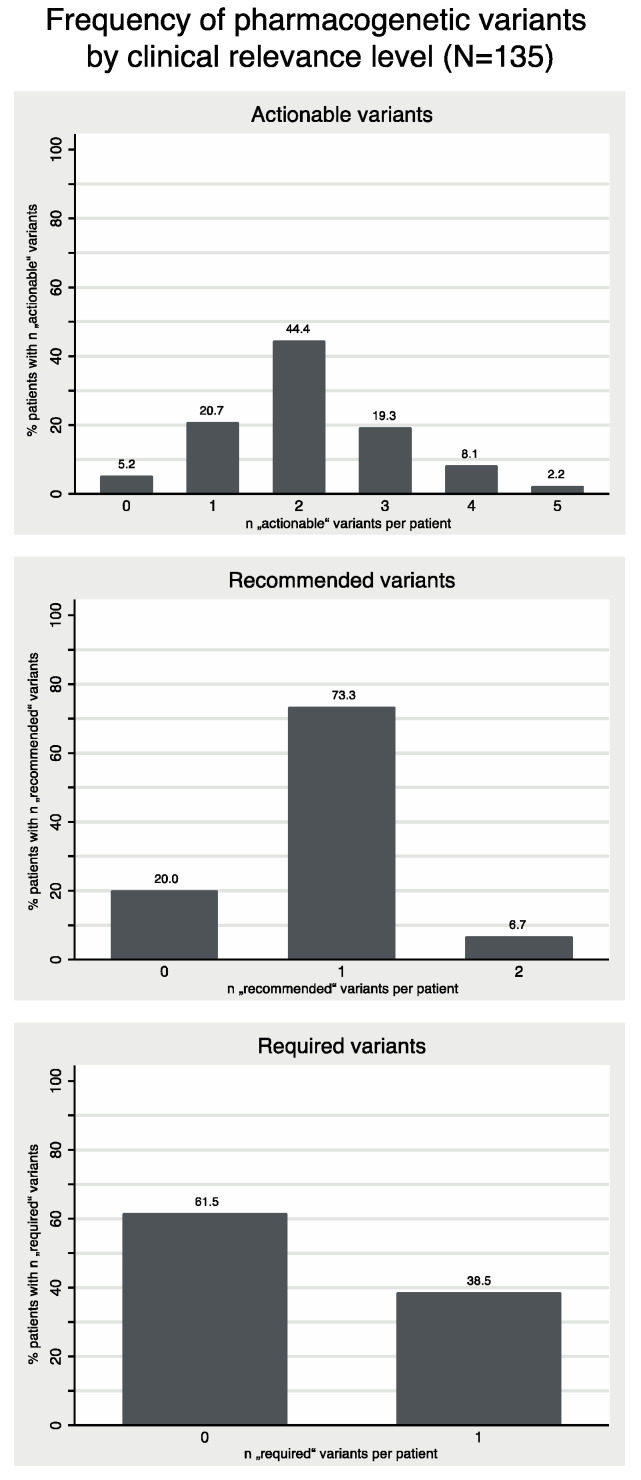
Distribution of number of variants per patient for “actionable”, “recommended” and “required” pharmacogenetic variants.

**Table 1 jcm-10-03200-t001:** Patient Characteristics.

	All Patients with PGx Panel Testing *n* (%)	Patients with Specific Drug–Gene-Based Indication *n* (%)	Patients with PGx Screening *n* (%)
*n* (%)	135 (100)	102 (75.5)	33 (24.5)
Age: median (range)	68 (25–92)	70 (25–92)	58 (30–83)
<60	48 (35.6)	30 (29.4)	18 (54.6)
61–70	25 (18.5)	22 (21.6)	3 (9.1)
71–80	41 (30.4)	33 (32.4)	8 (24.2)
>80	21 (15.6)	17 (16.7)	4 (12.1)
Sex			
male	81 (60)	56 (54.9)	25 (75.8)
female	54 (40)	46 (45.1)	8 (24.2)
eGFR < 60 mL/min ^1^	19 (14.1)	16 (15.7)	3 (9.1)
Main diagnosis			
Vascular disease	68 (50.4)	68 (66.7)	n.a.
Oncological disease	17 (12.6)	17 (16.7)	n.a.
Psychiatric disease	9 (6.7)	9 (8.8)	n.a.
Pain in orthopedic disease	7 (5.2)	7 (6.9)	n.a.
Gastric disease	1 (0.7)	1 (1.0)	n.a.
Indication PGx panel test			
Clopidogrel	60 (44.4)	60 (58.8)	n.a.
Tamoxifen	15 (11.1)	15 (14.7)	n.a.
Polypsychopharmacotherapy	9 (6.7)	9 (8.8)	n.a.
Opioids	7 (5.2)	7 (6.9)	n.a.
Statins	6 (4.4)	6 (5.9)	n.a.
Phenprocoumon	2 (1.5)	2 (2.0)	n.a.
Chemotherapy	2 (1.5)	2 (2.0)	n.a.
Proton pump inhibitor	1 (0.7)	1 (1.0)	n.a.
Pharmacotherapy			
Number of drugs, median (range) ^2^	6 (0–19)	6 (0–19)	3 (0–14)
Aspirin	43 (31.9)	38 (37.3)	5 (15.2)
Clopidogrel	48 (35.6)	48 (47.1)	0 (0)
Prasugrel or Ticagrelor	1 (0.7)	1 (1.0)	0 (0)
Coumarines or NOAC	25 (18.5)	22 (21.6)	3 (9.1)
Beta blockers	44 (32.6)	35 (34.3)	9 (27.3)
ACE inhibitors or ARB) ^3^	60 (44.4)	48 (47.1)	12 (36.4)
Calcium channel blockers	20 (14.8)	16 (15.7)	4 (12.1)
Diuretics	34 (25.2)	28 (27.5)	6 (18.2)
PPI	45 (33.3)	40 (39.2)	5 (15.2)
Cholesterol lowering drugs	55 (40.7)	48 (47.1)	7 (21.2)
NSAR	12 (8.9)	11 (10.8)	1 (3.0)
Opioids	17 (12.6)	14 (13.7)	3 (9.1)
Uric acid lowering drugs	5 (3.7)	3 (2.9)	2 (6.1)
Benzodiazepines	18 (13.3)	14 (13.7)	4 (12.1)
Antidepressants	28 (20.7)	24 (23.5)	4 (12.1)
Antipsychotics	10 (7.4)	9 (8.8)	1 (3.0)
Antiepileptics	9 (6.7)	8 (7.8)	1 (3.0)
Antidiabetics	22 (16.3)	17 (16.7)	5 (15.2)
Tamoxifen	12 (8.9)	12 (11.8)	0 (0)
CYP2C19 Inhibitor ^4^	11 (8.2)	10 (9.8)	1 (3.0)
CYP2D6 Inhibitor ^4^	26 (19.3)	20 (19.6)	6 (18.2)

^1^ eGFR calculated by using CKD-EPI formula (Levey et al., Ann Intern Med 2009, 150(9), 604-12); no data available for 58 patients. ^2^ One patient with indication of tamoxifen did not take any drugs at the time of PGx testing. ^3^ ACE = angiotensin converting enzyme inhibitors, ARB = angiotensin renin blockers. ^4^ Patients with at least one inhibitor, list of considered CYP2C19 inhibitors according to mediQ provided in Appendix A.

**Table 2 jcm-10-03200-t002:** Genes tested with SONOGEN panel, PGx levels and detected genetic variants.

Gene	Drugs with *Required* PGx-Testing ^1^	Drugs with *Recommended* PGx-Testing ^1^	Drugs with *Actionable* PGx-Testing ^1^	*n* (%) Patients with Phenotype Variants ^2^
*ABCB1*	-	-	-	106 (78.5) ^4^
***CYP2C9***	siponimod	-	celecoxib, phenytoin, warfarin	46 (34)
***CYP2C19***	-	atazanavir	amitriptyline, carisoprodol, citalopram, clobazam, clomipramine, clopidogrel, desipramine, doxepin, imipramine, nortriptyline, pantoprazole, trimipramine, voriconazole	71 (52)
***CYP2D6***	pimozide, tetrabenazine	-	amitriptyline, aripiprazole, atomoxetine, brexpiprazole, carvedilol, cevimeline, citalopram, clomipramine, clozapine, codeine, darifenacin, desipramine, doxepin, fesoterodine, iloperodine, nortriptyline, perphenazine, propafenone, tamoxifen, thioridazine, tramadol, trimipramine, vortioxetine	67 (50)
*SLCO1B1* ^3^	-	-	-	30 (22)
*VKORC1*	-	-	warfarin	86 (63.7)
*COMT*	-	-	-	73 (54.1)
*CYP1A2*	-	-	-	65 (48.6)
*CYP2B6*	-	-	efavirenz	67 (49.6)
*CYP3A4*	-	-	codeine, tamoxifen	6 (4.4)
*CYP3A5*	-	-	-	17 (12.6)
*CYP4F2*	-	-	warfarin	66 (48,9) ^4^
*DPYD* ^4^	-	-	capecitabine, fluorouracil	5 (3.7)
*OPRM1*	-	-	codeine	34 (25.2)
*POR*	-	-	-	72 (53.3) ^4^
***TPMT***	-	azathioprine, mercaptopurine	tioguanine	8 (5.9)

^1^ PGx level of drug–gene pairs according to PharmGKB, **genes in bold** feature at least one corresponding drug with a PGx level of required/recommended, informative not listed. ^2^ Variant phenotype = “non-normal” phenotype according to PharmGKB, not all variants are clinically relevant. ^3^ PGx level has been changed to “actionable” by FDA for rosuvastatin and to “recommended” by Swissmedic for simvastatin during the course of the study. ^4^ PGx level has been changed to “recommended” by EMA for capecitabine/fluorouracil during the course of the study.

**Table 3 jcm-10-03200-t003:** Drugs triggering PGx-testing, detected phenotype variants and recommendations to change patients’ current medication.

*n*	Drugs that Triggered PGx-Testing	Relevant Gene(s)	Detected Phenotype Variants ^1^	Patients with SONOGEN XP Recommendation to Change Triggering Drug	Patients with Clinical Expert Recommendation to Change Triggering Drug	Patients with Additional Clinical Expert Recommendations for Current but Nontriggering Drug(s) ^2^
**102**	**All patients with specific indication**	n.a.	n.a.	36 (35.3%)	33 (32.4%)	23 (22.5%)
**60**	Clopidogrel	*CYP2C19*	**1 PM/19 IM/** 19 RM or UM	20 (33.3%)	19 (31.6%)	16 (26.7%)
**15**	Tamoxifen	*CYP2D6*	**3 IM**/1 UM	3 (20.0%)	1 (6,7%)	0
**9**	Polypsycho-pharmacotherapy	*CYP1A2* *CYP2D6* *CYP2C19*	1A2: **6 UM** *CYP2D6*: **7 IM/1 PM** *CYP2C19*: **6 UM**	5 (55.6%)	3 (33.3%)	3 (33.3%)
**7**	Opioids	*OPRM CYP2D6*	*OPRM1* 3 **decreased function** CYP2D6: **4 IM**	4 (57.1%)	3 (42.9%)	4 (57.1%)
**6**	Statins	*SLCO1B1*	4 decreased or poor function	2 (33.3%)	4 (66.7%)	0 (%)
**2**	Phenprocoumon	*VKORC1 CYP4F2 CYP2C9*	***VKORC:* 1 normal, 1 decreased function** ***CYP2C9*: 2 normal function** ***CYP4F2*: 2 normal function**	1 (50.0%)	2 (100%)	0
**2**	Chemotherapy	*DPYD*	0	0	0	0
**1**	Proton pump inhibitor	*CYP2C19*	**1 UM**	1 (100%)	1 (100%)	0
**33**	**Screening**	n.a.	n.a.	n.a.	n.a.	3 (9.1%)

n.a. = not applicable (no triggering drugs in screening patients), NM = normal metabolizer, IM = intermediate metabolizer, PM = poor metabolizer, RM = rapid metabolizer, UM = ultrarapid metabolizer. ^1^ Variant = “non-normal” phenotype according to PharmGKB, **phenotypes in bold** = clinically relevant for triggering drug(s). ^2^ Based on PGx results, related drug–gene pairs are listed in Appendix A.

**Table 4 jcm-10-03200-t004:** Detected phenotype variants and related alerts relevant for potential future medication.

Trigger for PGx-Testing	n Patients	n Patients with ≥1 “Required” or “Recommended” PGx Variant ^1^	n SONOGEN XP Recommendations ^2^ per Patient Median (Range)	n Highlighted Clinical Expert Recommendations ^3^ per Patient Median (Range)
Specific PGx drug	**102**	88 (86.3%)	2 (2–11)	2 (0–6)
Screening	**33**	29 (87.9%)	5 (3–9)	3 (1–5)
**All Patients**	**135**	**117 (86.7%)**	**5 (2–11)**	**3 (0–6)**

^1^ Patients with at least one relevant phenotype variant for a gene featuring a PGx level of required or recommended on PharmGKB, i.e., IM or PM for *TPMT*, *CYP2C19* or *CYP2D6*. ^2^ Automatically generated, based on clinical annotations on PharmGKB. ^3^ Assessed as clinically relevant considering expert evaluation and individual patient history.

## Data Availability

The data that support the findings of this study are available on request from the corresponding author. The data are not publicly available due to privacy or ethical restrictions.

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
