# Peer review of "Clinical Relevance of a 16-Gene Pharmacogenetic Panel Test for Medication Management in a Cohort of 135 Patients"

_jcm, 2021, doi:10.3390/jcm10153200_

Round 1

Reviewer 1 Report

This well conceived and executed study gives a down to earthapplicatin of drug genetics to the real life situation. It is surprising how often clically relavant decisions originate from this basis.

As recommended by the authors a large scale outcome study would be a crowning achievement.

Reviewer 2 Report

Niedrig et al describe their experience from the implementation and interpretation of a PGx panel test and its impact in changes and recommendations on medication use and dosing. The design is an observational cohort study using a 16-gene PGx panel test, reports automatically generated by the SONOGEN XP expert system and personalised recommendations or change of medication.

The manuscript is interesting in describing a local experience using both a directed (specific drug-gene pair request) and a pre-emptive strategy of pharmacogenetics, however, the number of cases is low and mostly related to PGx of two drug, especially considering that the study has been performed in two tertiary care hospital for 2 years.

I have some questions and comments about the study

  • The authors should specify if the 135 patients included in the study were all the patients with a PGx test requested within the study timeframe or if some selection criteria were specified to select patients from all patient who have a PGx request by the attending physician. If the later, author should make an accountability of the process.
  • Please, add the most frequent main diagnosis of patients (although we can guess from the PGx indication).
  • Which were the criteria for a “screening” or pre-emptive indication. Is there any protocol or procedure for each situation?
  • All the recommendations were followed by real changes in the drugs (or drug dose)?
  • It is no clear to me the drugs included in the Pharmacotherapy section of table 1 (i.e., 60 patients in the Indication PGx panel test and 48 in the Pharmacotherapy section).
  • I think that it should be useful that authors provide the number of patients with genotypic variants that had also concomitant drugs potentially interfering with the drug eliciting the PGx test. This would give an idea of the complexity of the cases and the recommendations given.
  • In relation with the previous point, how the gene-DDI was managed? Any systematic procedure and/or source of evidence was used during the process of writing recommendations?
  • How many drugs taken by the patients had “required” or “recommended” genes not included in the 16-gen panel?
  • I think that readers would appreciate to know details about:
    • If any plan for PGx services were set-up and in that case, the main steps of the plan.
    • The process of test requesting: services demanding test and if previous protocols have been agreed with them. Reasons for PGx screening, etc.
    • Are patients attended directly or recommendation are made based on the clinical records?. In my opinion, it is interesting you describe your procedures and experience in setting up the PGx consultation and the interaction with patients and other clinicians. Much of your finding are similar to other groups, but the way you address the implementation of PGx in your hospitals can be useful to others.

Round 2

Reviewer 2 Report

I have no further questions. In my view, the manuscript have some interest related to the experience on implementation of pharmacogenetics (something that is difficult in many  clinical enviroments) but still it has moderate originality and the relevance is limited by the low numbers and no description of outcomes.

This manuscript is a resubmission of an earlier submission. The following is a list of the peer review reports and author responses from that submission.